# A Plant-Produced Porcine Parvovirus 1-82 VP2 Subunit Vaccine Protects Pregnant Sows against Challenge with a Genetically Heterologous PPV1 Strain

**DOI:** 10.3390/vaccines11010054

**Published:** 2022-12-26

**Authors:** Kyou-Nam Cho, In-Ohk Ouh, Young-Min Park, Min-Hee Park, Kyung-Min Min, Hyang-Ju Kang, Su-Yeong Yun, Jae-Young Song, Bang-Hun Hyun, Choi-Kyu Park, Bo-Hwa Choi, Yoon-Hee Lee

**Affiliations:** 1BioApplications Inc., Pohang-si 37668, Gyeongsangbuk-do, Republic of Korea; 2Viral Disease Division, Animal and Plant Quarantine Agency, Gimcheon-si 39660, Gyeongsangbuk-do, Republic of Korea; 3BioPOA, Hwaseong-si 18469, Gyeonggi-do, Republic of Korea; 4College of Veterinary Medicine, Kyungpook National University, Daegu-si 41566, Gyeongsang-do, Republic of Korea

**Keywords:** porcine parvovirus 1, plant-produced subunit vaccine, virus-like particle, viral protein 2, cross reactivation

## Abstract

Porcine parvovirus (PPV) causes reproductive failure in sows, and vaccination remains the most effective means of preventing infection. The NADL-2 strain has been used as a vaccine for ~50 years; however, it does not protect animals against genetically heterologous PPV strains. Thus, new effective and safe vaccines are needed. In this study, we aimed to identify novel PPV1 strains, and to develop PPV1 subunit vaccines. We isolated and sequenced PPV1 *VP2* genes from 926 pigs and identified ten PPV1 strains (belonging to Groups C, D and E). We selected the Group D PPV1-82 strain as a vaccine candidate because it was close to the highly pathogenic 27a strain. The PPV1-82 VP2 protein was produced in *Nicotiana benthamiana.* It formed virus-like particles and exhibited a 2^11^ agglutination value. The PPV1-190313 strain (Group E), isolated from an aborted fetus, was used as the challenging strain because it was pathogenic. The unvaccinated sow miscarried at 8 days postchallenge, and mummified fetuses were all *PPV1*-positive. By contrast, pregnant sows vaccinated with PPV1-82 VP2 had 9–11 Log_2_ antibody titers and produced normal fetuses after PPV1-190313 challenge. These results suggest the PPV1-82 VP2 subunit vaccine protects pregnant sows against a genetically heterologous PPV1 strain by inducing neutralizing antibodies.

## 1. Introduction

Porcine parvovirus (PPV) was first recognized as the causative agent of the stillbirths, mummification, embryonic death, and infertility (SMEDI) syndrome at the end of the 1960s [1,2]. Since then, PPV has spread widely and become a major cause of reproductive failure in susceptible pigs [3,4]. The PPV genome has two open reading frames, which give rise to genes encoding the nonstructural (NS) and viral (VP) proteins. VP2 is the main target of neutralizing antibodies against PPV [3]. Various molecular methods developed in the last two decades have allowed the identification of new PPVs in pig herds worldwide. According to the International Committee on the Taxonomy of Viruses (ICTV) classification, to date, seven genotypes of PPV have been discovered in pigs; these belong to four genera: *Protoparvovirus* (PPV1), *Tetraparvovirus* (PPV2–3), *Copiparvovirus* (PPV4–6) and *Chapparvovirus* (PPV7), based on their NS1 homologies [5,6].

In Republic of Korea, studies aiming to detect and characterize different PPV strains began in 2001 [7]. Of the 481 wild boar sera tested between 2010 and 2011, 29.5% (142/481) were seropositive for PPV antibodies [8]. The T142 strain isolated in Republic of Korea from the domestic pig is identical to the Kresse strain isolated in the USA in 1985 [9]. In 2018, Chung et al. identified seven PPV1 and two PPV7 strains among the 151 samples isolated from domestic pig tissues [10]. Moreover, a South Korean study, which aimed to assess the genetic diversity of PPV in 202 wild boar samples between 2017 and 2018, found that 31 of the samples were positive for PPV (specifically 11 PPV1, 5 PPV3, 3 PPV4, 6 PPV5, 5 PPV6 and 1 PPV7) [11]. The inactivated VRI-1 strain (GenBank Accession Number: AY390557) belonging to the NADL-2 strain has been used in Republic of Korea as a porcine parvovirus vaccine. Despite vaccination with the VRI-1 strain, recurrence of PPV infections and virus-associated miscarriages and stillbirths has been reported [9,12].

Substitution mutations in the PPV sequence mainly target the capsid VP, therefore influencing its receptor binding and/or antigenicity [6]. The currently used PPV vaccines are based on the NADL-2 (Group A) strain, isolated ~50 years ago. These vaccines are effective against homologous infections but do not prevent infection and virus shedding after challenge with the antigenically heterologous 27a strain (Group D) [13]. Furthermore, infection of pigs or rabbits with the 27a strain induced 100–1000-fold lower homologous neutralizing antibody titers than the heterologous antibody titers elicited by the 143a, NADL-2, or IDT strains [14]. Vaccination with the inactivated 27a strain prevented fetal death after homologous viral challenge with live 27a. However, a substantial increase in antibody titers was detected after infection, indicating the presence of viral replication in the immunized pigs [15]. These experiments suggest that although current vaccines are able to prevent disease, inactivated vaccines cannot induce the desired level of sterile immunity against 27a [15]. The 27a strain may therefore have unique immunological features, which enable it to utilize a yet-undefined immune escape mechanism. 

Plants have been used as manufacturing hosts for treatments and subunit vaccines for more than 25 years [16,17]. Transient protein expression in *Nicotiana benthamiana* is an important platform for producing plant-made pharmaceuticals [18]. This platform is well-adjusted to the manufacture of veterinary goods because (1) it can be used to produce a wide range of products, (2) it produces materials at a lower cost than traditional cell culture-based expression platforms, (3) the equipment required is straightforward, and (4) production times are relatively short [19,20,21].

Since existing PPV vaccines do not offer enough protection against genetically heterologous strains, novel PPV vaccines are urgently required. The aims of this study were as follows: (1) to identify novel PPV1 strains by isolating and characterizing PPVs from pigs in South Korea, and (2) to develop an effective and safe PPV1 vaccine using a plant expression system.

## 2. Materials and Methods


**Sample collection**


Between 2017 and 2019, a total of 926 Korean domestic pig samples were collected from 215 farms and submitted to the Viral Disease Division of the Animal and Plant Quarantine Agency for the diagnosis of porcine reproductive failure disease; the samples comprised 272 aborted mixed tissue samples from pig fetuses (from 136 farms) and 654 lung tissue samples (from 79 farms).


**DNA extraction**


Samples were homogenized using the Precellys® CK28-R Lysing Kit and the Evolution homogenizer (Bertin Technologies, Montigny-le-Bretonneux, France). Total DNA was extracted using a commercial DNeasy Mini Kit (Qiagen, Germantown, MD, USA), following the manufacturer’s instructions.


**DNA sequencing and phylogenetic analysis**


Sequencing of near-full-length major capsid genes was conducted to evaluate genetic differences between PPV strains. The target gene was amplified using the amplification sequencing primers *PPV1* F 5’-ATGAGTGAAAATGTGGAACAA-3’ and *PPV1* R 5’-AGTATAATTTTCTTGGTATAAGTTGTGAATGTTC-3’. Positive amplicons were purified using an agarose gel extraction kit (Qiagen, Germantown, MD, USA) and ligated into the pDrive vector (Qiagen, Germantown, MD, USA), following the manufacturer’s instructions. Ligation products were transformed into *Escherichia coli* DH5 α-competent cells and incubated at 37 °C overnight. The constructed vectors were extracted using the Plasmid Miniprep Kit (Inclone, Sungnam, Republic of Korea) and sequenced using Macrogen vector-sequencing primers (Macrogen, Seoul, Republic of Korea). The obtained sequences were identified using the National Center for Biotechnology Information (NCBI) Basic Local Alignment Search Tool (BLAST). Multiple sequence alignments and nucleotide sequences homology searches of PPV1 isolates were performed using the CLC Main Workbench package (ver. 7.0.3; CLC Bio, Aarhus, Denmark). The sequence alignment results were modified using BioEdit ver. 7.2.5, accessed on 1 August 2020, https://bioedit.software.informer.com/7.2/) and analyzed using a similarity matrix. Phylogenetic analysis was performed using MEGA ver. 6.0 (https://megasoftware.net/), with reference sequences of major capsid genes obtained from the GenBank database; the maximum-likelihood approach and 1000 bootstraps replicate values were used.


**Cloning of the PPV1-82 *VP2* gene**


The sequence of the PPV1-82 *VP2* gene was deposited into the NCBI database (GenBank Accession Number; JQ249923.1) and optimized for expression in *Nicotiana benthamiana* (https://zendto.bioneer.co.kr/codon/index.py), prior to gene synthesis. The PPV1-82 *VP2* sequence was then cloned into the pCAMBIA1300 vector [22] via the *XbaI* and *ScaI* restriction sites. A vector for targeted expression in chloroplasts was constructed by sequential fusion of sequences encoding the rubisco transit peptide (Rbc), six histidines, and PPV1-82 *VP2* between the CaMV 35S promoter and the heat shock protein terminator of pCAMBIA1300. The nucleotide sequence was verified by sequencing (Bioneer, Daejeon, Republic of Korea).


**PPV1-82 VP2 purification**


The recombinant PPV1-82 *VP2* gene was transfected into the *Agrobacterium tumefaciens* LBA4404 strain (Takara, Kusatsu, Japan) by electroporation. Transformed *A. tumefaciens* were grown for 16 h in 5 mL of yeast extract peptone (YEP) liquid medium, supplemented with 50 mg/L kanamycin and 25 mg/L rifampicin. Next, 1 mL of cultured bacteria was inoculated into 50 mL of fresh YEP medium and cultured for a further 16 h at 28 °C. Bacteria were then pelleted by centrifugation at 7341× *g* for 5 min at 4 °C and resuspended at the desired concentration (determined by measuring OD_600_) in a solution consisting of 10 mM 2-(N-morpholino) ethane sulfonic acid (Duksan, Ansan, Republic of Korea), 10 mM magnesium chloride (Sigma-Aldrich, St. Louis, MI, USA), and 100 mM acetosyringone (Sigma-Aldrich, St. Louis, USA), at pH 5.6. Agroinfiltration was carried out by injecting the bacterial suspension into the underside of the *N. benthamiana* leaves using a needleless syringe. After 3 days, the leaves were harvested for protein purification. The leaves were then crushed and incubated for 30 min in an extraction buffer consisting of 25 mM sodium phosphate pH 8.0, 150 mM sodium chloride, 0.5% (*v*/*v*) Triton X-100, 100 mM sodium sulfite, and 1.5% polyvinylpolypyrrolidone. After centrifugation, the supernatant pH was adjusted to 5.3 with 14.3 M acetic acid, prior to heating it at 41 °C for 15 min. The supernatant was then neutralized to pH 6.5 with 1 M Tris and 1% diatomite for 20 min, treated with 0.0125% formaldehyde for 10 min, and then precipitated with 20% ammonium sulfate. The precipitate was loaded onto a HiLoad 16/600 Superdex 200 pg column (Sigma-Aldrich, St. Louis, USA) and an Increase 10/30 column of a fast protein liquid chromatography system (FPLC; AKTA purifier, GE Healthcare Systems, Chicago, IL, USA) at 4 °C in an elution buffer consisting of 25 mM sodium phosphate (pH 8.0) and 150 mM sodium chloride. Proteins were eluted in an elution buffer at a flow rate of 1 mL/minute, and the protein concentration was measured at 280 nm. Fractions corresponding to the PPV1-82 VP2 elution peak were collected, pooled, and used in further tests.


**Transmission electron microscopy (TEM)**


The purified PPV1-82 VP2 protein was diluted to a final concentration of 0.05 mg/mL in phosphate-buffered saline (PBS) for negative staining. A 5 μL aliquot of diluted protein was applied to the carbon-coated grid, which had previously been glow-discharged (Harrick Plasma, Ithaca, NY, USA) for 3 min in air, followed by negative staining with 1% uranyl acetate [23]. The prepared grids were examined by TEM on a JEM-2100 F microscope (Jeol, Akishima, Japan) operated at 200 kV. Images were acquired with an Ultra Scan 1000 CCD camera (Gatan, Pleasanton, CA, USA).


**Animal experiments and ethical approval**


Animal experiments were reviewed, approved, and supervised by the Institutional Review Board of Opti Pharm Inc. (Approval number: OPTI-IAC-2001). Biopsies of pregnant sows and fetuses were performed by the Animal and Plant Quarantine Agency. 


**Vaccination**


After acclimation for 1 week, sows were intramuscularly vaccinated with the PPV1-82 VP2 antigen (HA value = 2^13^) combined with the MONTANIDE IMS1313 adjuvant at 50% (*v*/*v*) (Seppic, La Garenne Colombes Cedex, France) at 4 and 2 weeks before conception.


**Immunogenicity**


Bloods were collected before vaccination and 4 weeks after vaccination, and used for hemagglutination inhibition analysis.


**PPV1-190313 strain challenge**


Porcine kidney (PK)-15 cells, cultured in Eagle’s Minimum Essential Medium (EMEM), supplemented with 10% (*v*/*v*) fetal bovine serum (FBS), 200 units/mL penicillin, and 100 μg/mL streptomycin, were infected with the PPV1-190313 strain. Cell cultures were observed for 5–6 days postinfection. The supernatant was collected after three freeze–thaw cycles and stored in a –80 °C deep freezer before reinfection. In the PPV1-190313 pathogenicity experiments, sows were infected with 8 mL of the PPV1-190313 strain (at 4.0 × 10^5^ median tissue culture infectious dose [TCID_50_]/mL); 4 mL was administered by intramuscular injection into the neck, and the other 4 mL was delivered intranasally via a spray. In the PPV1-82 VP2 vaccination experiments, the vaccinated sows were challenged with 4 mL of the PPV1-190313 strain (at 4.0 × 10^5^ TCID_50_/mL); 2 mL was administered by intramuscular injection into the neck, and the other 2 mL was delivered intranasally via a spray.


**Tissue sample collection**


In the PPV1-190313 pathogenicity experiments, blood, nasal fluid, and stool samples were collected from the sows at 0, 3, 5, 7, 14, 21, 28, 35, 42 and 49 days postchallenge (DPC). In the PPV1-82 VP2 vaccination experiments, blood, nasal fluid, and stool samples were collected from vaccinated sows at 0, 3, 5, 7, 14, 21, 28, 35 and 42 DPC. Samples were stored in a –80 °C deep freezer before analysis. At 53 DPC, sows were euthanized, and cesarean sections were performed to remove the fetuses. Submandibular lymph nodes, tonsils, hearts, lungs, livers, kidneys, spleens, uteri, and brains were collected from the sows. Lysate supernatant from the mummified and normal fetuses was also collected for analysis.


**Hemagglutination (HA) assay**


The HA test was performed on supernatants obtained after centrifugation of the tissue samples at 1000× *g* for 10 min at 4 °C. The 50 μL test samples were serially diluted two-fold in 50 μL of 0.1% (*v*/*v*) bovine serum albumin (BSA) in PBS and mixed in V-shaped microtiter plates. An amount of 50 μL of guinea pig red blood cells was then added to the wells. Positive and negative controls were included. After incubation for 45 min at room temperature, plates were examined for agglutination of the red blood cells. The antigen content was expressed as Log_2_ HA unit per 50 μL.


**Hemagglutination inhibition (HI) assay**


A two-fold serial dilution of 50 μL serum samples was prepared in microtiter plates and mixed with an equal volume of a solution containing 4 HA units of PPV1. After incubation for 45 min at room temperature, guinea pig red blood cells were added at a final concentration of 0.33% (*v*/*v*). After incubation overnight at 2–8 °C, plates were examined for red blood cell agglutination. The titer of antibodies inhibiting the PPV1-mediated agglutination of the red blood cells was expressed as Log_2_.


**PPV1-specific conventional PCR**


To detect *PPV1* gene presence, primers *PPV1* DF 5’-ACCAACATACACTGGACAATCAC-3’ and *PPV1* DR 5’- ATAGCACCATTTGGTTCATCA-3’ were designed to cover 458 base pairs of the *PPV1* gene. The PCR program included initial denaturation at 94 °C for 5 min, followed by 40 cycles at 94 °C for 30 s, 55 °C for 40 s, 72 °C for 30 s, and a final extension at 72 °C for 5 min. PCR products were separated by electrophoresis on a 2% agarose gel.


**Enzyme-linked immune sorbent assay (ELISA)**


Porcine serum samples were assayed for the presence of anti-PPV antibodies using the INgezim PPV Compac ELISA Kit (Ingenasa, Madrid, Spain), according to the manufacturer’s instructions. This is an enzymatic assay (based on a double antibody sandwich ELISA technique) that uses a monoclonal antibody specific for the PPV VP2 protein. Two blocking percentage (BP) values were used for result interpretation: samples with a BP higher than 30% were considered positive, and those with a BP lower than 25% were considered negative. Samples with BPs of 25–30% were considered borderline.


**Histological analysis**


Uteri and placentas were fixed in 4% paraformaldehyde solution for 24 h, followed by paraffin embedding and slicing into 4 μm sections. The sections were first stained with undiluted hematoxylin for 3–8 min, washed with tap water, differentiated with 1% hydrochloric acid alcohol for 5–10 s, and washed again with tap water. The cell nuclei were then stained with 0.6% ammonia water, and a red/pink to blue color change was observed. After a final wash in running water, the sections were placed in eosin staining solution for 1–3 min for cytoplasmic staining. Sections were viewed on an Olympus IX81 upright light microscope (Olympus, Shinjuku, Japan).


**Serum Neutralization**


Porcine Kidney-15 (PK-15) cells’ monolayers were prepared in the microplates by seeding 0.2 mL of cell suspension in growth EMEM containing 2 × 10^5^ cells/mL, and incubated at 37 °C for 18–24 h. Sera were inactivated at 56 °C for 30 min and then diluted two-fold in DMEM in a 96-well flat-bottomed tissue culture plates (Nunc, Rochester, NY, USA). Virus suspension with a titer of 100 TCID_50_ in 50 μL was added to each serum sample and then incubated for 1 h at 37 °C and 5% CO2. PK-15 cell suspension (50 μL) was added to each well and incubated for 3–7 days. Appropriate serum, virus, and cell controls were included in this test. The cells were monitored for cytopathic effects (CPE) by light microscopy.


**Statistical analysis**


Statistical analysis was performed using the Student’s unpaired t-test, with statistical significance set at * *p* < 0.05; ** *p* < 0.01.

## 3. Results

### 3.1. PPV1-82 Strain Selection

We first analyzed the prevalence of PPV1 in porcine samples over 2.5 years, from 2017 to 2019. We sequenced the PPV1 VP2 gene in 926 domestic pig samples and identified ten different PPV1 strains, which were named 13, 20, 21, 22, 23, 25, 49, 82, 1220 and 190313. Nine of the strains were obtained from abattoir samples, and one strain (190313) originated from an aborted pig fetus.

The isolated PPV strains were categorized into Groups C, D and E (Figure 1). PPV1-82 was placed in Group D, with the highly pathogenic 27a strain isolated in Germany (GenBank Accession Number: AY684871). PPV1-49 and PPV1-25 were placed in Group C. The other seven strains were placed in Group E, together with the T142 strain isolated in Republic of Korea in 2017.

We found that the VP2 sequence of PPV1-82 contained the same mutations at amino acid positions 45, 215, 228, 278, 383, 414, 419, 436 and 565 as the VP2 of the 27a strain. However, both PPV1-82 and 27a VP2 sequences had completely different mutations at these sites than the NADL-2 vaccine strain in Group A (Appendix A). The NADL-2 strain was first isolated in the United States in 1972 [24], and the inactivated NADL-2 vaccine has been used for ~50 years.

### 3.2. PPV1-82 VP2 Protein Purification and Immunogenicity

The emergence of novel PPV1 strains should be mitigated by the development of new PPV1 vaccines. In this study, we selected the PPV1-82 strain as a candidate for new PPV1 subunit vaccine development. We purified the PPV1-82 VP2 protein in *N. benthamiana* by size-exclusion chromatography (Figure 2A). Protein separation was carried out using a Nickel-affinity column, but target proteins did not bind at all, and ammonium sulfate precipitation was used. The purified PPV1-82 VP2 antigen was not completely pure, because of three major bands. In the immunostaining of plant extract after transient transfection, the upper two bands were detected by His-monoclonal antibody (data not shown). The purified PPV1-82 VP2 protein in fractions 32–36 was used to generate virus-like particles (VLP) with diameters ranging from 22.3 to 25.7 nm (Figure 2B). The HA value for PPV1-82 VP2 VLP was 2^11^ (Figure 2C). To determine the immunogenicity of the PPV1-82 VP2, we vaccinated three piglets with the PPV1-82 VP2 antigen (HA value = 2^13^), combined with the MONTANIDE IMS1313 adjuvant (50% [*v*/*v*]). An amount of 2 mL of the vaccine was injected intramuscularly per piglet. The piglets’ background was Crossbred Pig (Landrace × Yorkshire × Duroc), and 12–13-week-old piglets that were negative to *PPV1* gene-specific PCR with low serum HI titer were selected. Blood samples were collected before vaccination and 4 weeks after vaccination. The HI units in the sera of vaccinated piglets (about 10 Log_2_) were significantly higher at 4 weeks postvaccination than the HI units of piglets at week 0 or those of unvaccinated animals (Figure 2D).

### 3.3. Pathogenicity of the PPV1-190313 Strain

In this study, we isolated one PPV1 strain from an aborted fetus and labeled it PPV1-190313 (GenBank Accession Number: MZ856459). The PPV1-190313 strain was placed in Group E, together with the T142 strain isolated in Republic of Korea in 2017 [9]. After 40 passages in PK-15 cells, we measured the titer of the PPV1-190313 strain and obtained an HA value of 2^7^ (Figure 3A). We examined the pathogenicity of the PPV1-190313 strain using three Specific Pathogen-Free (SPF) Yukatan miniature once-pregnant sows (W19-090, W19-117 and W19-118) with no prior PPV, porcine reproductive and respiratory syndrome virus (PRRSV), porcine circovirus2 (PCV2), or classical swine fever virus (CSFV) infections. We inoculated each sow with 8 mL of PPV1-190313 on gestation day 40; 4 mL of 4 × 10^5^ TCID_50_/mL of PPV1-190313 was injected intramuscularly into the neck, and 4 mL was delivered intranasally. The W19-090 sow showed signs of clinical anorexia and then aborted four fetuses at 9 days postchallenge (DPC). Whole blood and nasal fluid samples were collected from sow W19-090 at 0, 3, 5, 7, 14, 21, 28, 35, 42 and 49 DPC, and conventional PCRs were performed. Sows W19-090 and W19-117 had PPV1 in the blood and nasal fluid until 49 DPC (Table 1). However, sow W19-118 had a PPV1-positive nasal fluid sample but a PPV1-negative blood sample at 49 DPC. Sows were euthanized at 53 DPC, and the fetuses were removed from sows W19-117 and W19-118 by cesarean section. We detected the PPV1 gene in the tissues (including the submandibular lymph node, brain, spine, lung, heart, liver, kidney, spleen, uterus, and tonsil) of all three sows. The histology results revealed that the sow that had miscarried (W19-090) had a morphologically normal uterus; the hematoxylin and eosin staining of the uterus and placenta did not show any signs of lesions (Figure 3B). Sow W19-117 had a morphologically normal uterus; however, three of its six fetuses were mummified (Figure 3C). Examination of sow W19-118 revealed that all eight of its fetuses were mummified (Figure 3D). While the tissues from non-mummified fetuses were sporadically positive for PPV1 (Table 2), those of mummified fetuses were all positive for PPV1 (data not shown). The HI units and neutralizing antibodies in sows challenged with PPV1-190313 were detected at 5 DPC. The HI units were maintained at >12 Log_2_ throughout the experiment. The neutralizing antibody titers were also significantly elevated until 21 DPC and maintained at >10 Log_2_ until the end of the experiments (Appendix A).

### 3.4. PPV1-82 VP2 Vaccination Protects Pregnant Sows and Fetuses against the PPV1-190313 Strain

To confirm the protective effect of the PPV1-82 VP2 vaccine, three sows (labeled 1, 2 and 3) were vaccinated with 10^4^ HA of PPV1-82 VP2 adjuvanted with 50% (*v*/*v*) MONTANIDE IMS1313 at 4 and 2 weeks before conception. Because of the strong pathogenicity of the PPV-190313 strain, in these experiments, we used half of the viral titer used in the pathogenicity experiments. The sows were impregnated and then challenged with 4 mL of the PPV1-190313 strain containing 4.0 × 10^5^ TCID_50_/mL at gestation day 40; 2 mL was administered by intramuscular injection into the neck, and 2 mL was delivered intranasally via a spray. At 53 DPC, sows were euthanized, and cesarean sections were performed to remove the fetuses. Blood, nasal fluid, and stool samples were collected from the sows at 0, 3, 5, 7, 14, 21, 28, 35 and 49 DPC (Figure 4A). Vaccinated sows 1, 2 and 3 had six, four, and five nonmummified fetuses, respectively (Figure 4B). The unvaccinated PPV1-190313 control sow developed a cough after viral challenge and then miscarried eight fetuses at 8 DPC (Figure 4C). The unvaccinated and unchallenged negative control sow had six normal fetuses (Figure 4D). Blood, nasal fluid, and stool samples from vaccinated sows were largely negative for the PPV1 gene; only the stool sample of the vaccinated sow 2 was PPV1-positive at 3 DPC (Table 3). The blood, nasal fluid, and stool samples from the unvaccinated PPV1-190313 control sow were positive for the PPV1 gene until 7, 14 and 7 DPC, respectively (Table 3). The tissues from vaccinated sows were mostly negative for PPV1, except for the spleen of vaccinated sow 3. However, excluding the brain, the tissues of the unvaccinated PPV1-190313 control sow were positive for PPV1 at 53 DPC (Table 4). The HI units of vaccinated sows were 12–13 Log_2_ at 0 DPC, and these values were maintained in the 12–14 Log_2_ range until 49 DPC. In addition, the serum neutralization (SN) units of the vaccinated animals were 6–7 Log_2_ at 0 DPC and were sustained above 8 Log_2_ until 49 DPC. The HI unit of the unvaccinated PPV1-190313 challenge control was <2 Log_2_ at 0 DPC but later rose to >13 Log_2_, where it was maintained for the duration of the experiment; in this sow, the SN units increased after 14 DPC and were maintained at over 9 Log_2_ until 49 DPC (Appendix A).

## 4. Discussion

In this study, we evaluated the diversity of the *VP2* gene in ten PPV1 strains. Of the ten PPV1 strains studied, seven PPV1 strains were in Group E, one strain was in Group D, and two strains were in Group C (together with the T142 South Korean strain [GenBank Accession Number: KY994646]). The maximum clade credibility tree, constructed using the VP1 and VP2 sequences of the T142 strain, showed four major distinct lineages. Group 1 includes the European Challenge strain (GenBank Accession Number: AY684866.1), isolated in the United Kingdom in 1986. Group 2 contains the German vaccine strain IDT (GenBank Accession Number: AY684872.1). Group 3 comprises Asian strains, mostly from China, which are similar to the vaccine strain NADL-2 (GenBank Accession Number: NC_001718.1). Group 4 includes strains isolated in various European and Asian countries and the USA; these strains are similar to the Kresse strain (GenBank Accession Number: U44978.1), which was isolated in the USA in 1985 [9].

The PPV1-82 strain we had identified was closely related to the highly pathogenic 27a strain in Group D. PPV1-82 and 27a had the same mutations at amino acid positions 228, 419 and 436, and differed from the low pathogenic NADL-2 vaccine strain in Group A (Appendix A). The evolution mean rate for the PPV1 *VP2* gene is estimated to be 5.47 × 10^−5^ substitutions/site/year. Moreover, 28 highly variable *VP2* regions, which include amino acids 228, 419, and 436, were reported [9]. Zeeuw et al. found that the threonine (T) substitution at amino acid position 436 of VP2 was especially important in the virulence of 27a as it could potentially modify the pathogenicity of PPV in vivo [14]. The PPV1-82 VP2 subunit vaccine was sufficiently expected to respond against homogeneous Group D strains. In addition, we confirmed the cross-reactivity by PPV1-82 VP2 against the 190313 strain of heterogenous Group E.

Cross-neutralization studies conducted on the IDT and NADL-2 vaccine strains, which exhibited low neutralization activity against the 27a strain, indicate that current PPV1 vaccines are ineffective at preventing PPV1 spread [13,14]. Although these vaccines are effective against homologous infections, they do not prevent viral shedding and infection after challenge with the antigenically heterologous 27a strain. Infection of pigs or rabbits with the 27a strain induces homologous neutralizing antibody titers that are 100–1000-fold lower than the heterologous neutralizing antibody titers generated against the NADL-2, 143a, or IDT strains [13]. Alarmingly, the inactivated 27a, NADL-2, and IDT strains can infect fetuses via transplacental transmission [14]. The hope is that new PPV1 subunit vaccines would prevent this from happening.

The PPV capsid is a spherical shell of 60 identical copies of VP1, VP2 and VP3, arranged with icosahedral symmetry [25]. VP2, the major capsid structural protein, can elicit PPV-specific neutralizing antibodies [26,27]. VLPs remain a topic of considerable interest because of their ability to produce a range of immune responses [28], and vaccination is still the most effective means of preventing porcine parvovirus disease [13]. Attenuated viral vaccines are traditionally inactivated forms of the original pathogen and are produced using cell lines in vitro. Although these types of vaccines provide effective PPV control, safety concerns are often raised because of the risk of incomplete inactivation of the virus, which could lead to viral replication in the host [29]. A Chinese group reported that a VP2 VLP vaccine, produced by expressing the PPV1 JS strain in *E. coli* Transetta (DE3) competent cells, induced a 2^12^ HA titer with an HI of 8–9 Log_2_ in weaned pigs [30]. In our study, the PPV1-82 VP2 protein was produced using *N. benthamiana* plant cells and also formed VLPs. In piglets, the plant-produced PPV1-82 VP2 VLP had a 2^11^ HA value and an HI of 10 Log_2_ units. This suggests that the PPV1-82 VP2 VLPs could induce a similar immune response to that elicited by live PPV1-82 virions.

In this study, the PPV1 strain 190313 was isolated from a single aborted fetus, while the other PPV1 strains were isolated from mixed pig tissue samples. Because of its high pathogenicity, we decided to use PPV1-190313 as a challenge strain in our experiments. We found that the PPV1-190313-infected pregnant sows shed the virus throughout the pathogenicity experiment. Of the three infected sows, one miscarried at 9 DPC, and the others had mummified fetuses in utero. In these animals, PPV1-190313 had crossed the placental barrier, leading to congenital infection. Thus, PPV1-190313 is a highly virulent strain, which is vertically transmitted to the fetus. In the pathogenicity experiments, the infection dose of PPV1-190313 was 8 mL at 4.0 × 10^5^ TCID_50_/mL. Because this dose was too pathogenic, we decided to halve the dose in the subsequent vaccination and challenge experiments. Challenge with this new dose did not induce histological lesions in the uteri or placentas of the infected sows. 

The outcome of fetal infection varies with the progression of gestation. For instance, experimental and epidemiological studies indicate that PPV infection during the first half of pregnancy leads to reproductive failure [31,32]. The gestation period of sows is about 110 days. PPV1-190313 infection at gestation day 40 could therefore induce miscarriage. In this study, we observed that the anti-PPV1 neutralizing antibody titers rose in PPV1-190313-infected pregnant sows after 5 DPC and were maintained at over 10 Log_2_ after 21 DPC. However, we surmised that the fetuses had already been exposed to PPV1-190313 before a robust neutralizing antibody response could be mounted. The vaccination and challenge experiments, however, showed that vaccination with PPV1-82 VP2 protected the sows against infection with the PPV1-190313 strain. The blood, nasal fluid, and stool samples from vaccinated sows were negative for the *PPV1* antigen until 49 DPC. Moreover, the vaccinated sows produced only normal fetuses, while the unvaccinated PPV1-190313 control sow miscarried all eight fetuses. We found that the PPV1-190313 control sow was positive for the *PPV1* gene until 7 DPC, after which *PPV1* was not detected again until 14 DPC. In addition, the tissues isolated from the PPV1-190313 control sow were positive for the *PPV1* gene. These results suggest that the PPV1-82 VP2 subunit vaccine produced in plants protects sows against infection with the genetically heterologous PPV1-190313 strain.

Although PPVs may have a common ancestor, the high genomic variability among these viruses influences their pathogenic potential [33]. Unlike for PPV1, the pathogenic role of newly identified PPVs (PPV2–7) is not yet clearly defined. PPV2 has been isolated predominantly from lung samples and is associated with the onset of respiratory diseases [34,35] such as porcine respiratory disease complex, which is thought to implicate alveolar macrophages [36]. Kim et al. reported that porcine circovirus 2 (PCV2)-positive lung samples, with or without porcine reproductive and respiratory virus (PRRSV) positivity, had significantly higher titers of PPV1 and PPV6. In contrast, the prevalence of PPV2 and PPV7 was significantly higher in the lungs of PRRSV-infected pigs, regardless of PCV2 coinfection. PPV5, however, was detected significantly more frequently in pigs coinfected with either PCV2 or PRRSV [12]. We therefore next need to evaluate whether the PPV1-82 VP2 vaccine also cross-protects pigs against PPV2–7 infection in the context of porcine reproductive and respiratory diseases.

In conclusion, we have demonstrated that the plant-produced PPV1-82 VP2 forms VPLs and protects pregnant sows and their fetuses against infection with the genetically heterogenous PPV1 190313 strain. We anticipate that this novel PPV1 vaccine could be a promising PPV vaccine candidate. 

## Figures and Tables

**Figure 1 vaccines-11-00054-f001:**
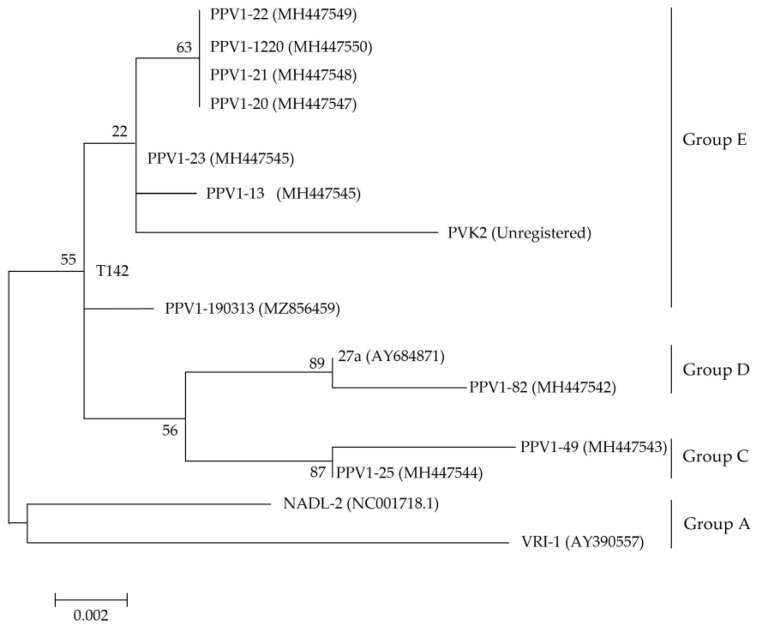
Phylogenetic analysis based on PPV1 *VP2* gene sequences. The maximum-likelihood method (Tamura-Nei model of nucleotide substitution, gamma distributed with invariant sites) was used in molecular epidemiology and analysis of rapidly evolving heterochronos PPV1 *VP2* sequences, with the 1000 bootstrap replicates setting of the MEGA v 6.0 program. VP2 amino acid sequences of the ten PPV1s isolated from Korean pig farms were compared with the reference sequences PVK2, T142, 27a, NADL-2 and VRI-1. GenBank Accession numbers are provided in the parentheses next to each strain name. The log likelihood (log L) value for the genome sequences was −28,543.99. Bootstrap values are indicated at the nodes. The scale bar indicates the number of nucleotide substitutions per site.

**Figure 2 vaccines-11-00054-f002:**
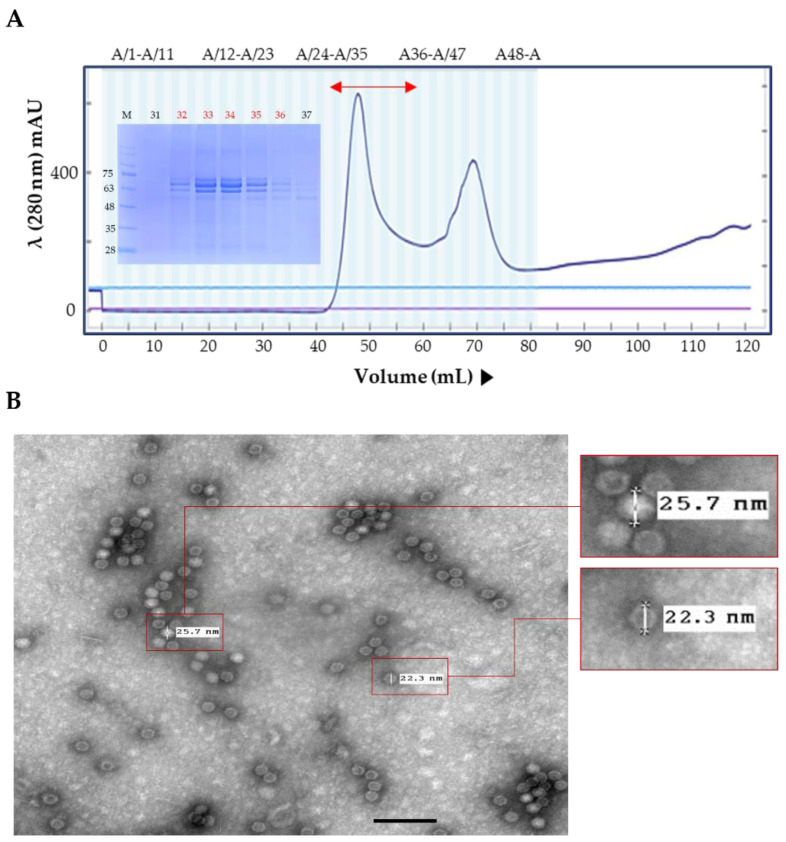
Plant-produced PPV1-82 VP2 protein purification and immunogenicity. (**A**) The PPV1-82 VP2 size-exclusion chromatography trace and an SDS-PAGE gel image of the collected fractions. The bidirectional red arrow indicates the elution fractions (32–36) that the purified PPV1-82 VP2 protein was collected from. (**B**) Transmission electron microscopy image of virus-like particles formed using the purified PPV1-82 VP2 protein (scale bar = 100 nm, HV = 75.0 kV, direct magnification: 100,000×). (**C**) HA results, sedimentary red blood cells represented non-agglutination. (**D**) Results from the HI assay performed on samples from three piglets vaccinated with PPV1-82 VP2; wpv, weeks postvaccination; * *p* < 0.05; ** *p* < 0.01.

**Figure 3 vaccines-11-00054-f003:**
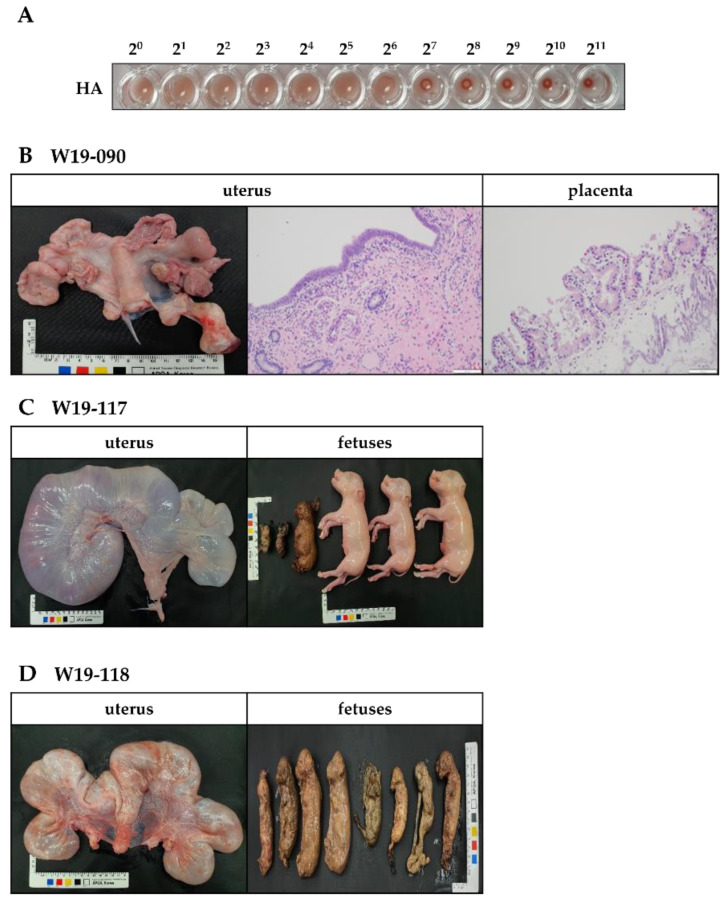
Pathogenicity of PPV1-190313 in three gestational Yukatan miniature sows. Sows were infected with 8 mL of the PPV1-190313 strain (at 4.0 × 10^5^ [TCID_50_]/mL); 4 mL was administered by intramuscular injection into the neck, and the other 4 mL was delivered intranasally via a spray. (**A**) HA assay results obtained with PPV1-190313 after 40 passages in porcine kidney-15 cells. (**B**) Photographic and hematoxylin and eosin staining images of the uterus from the W19-090 sow, which miscarried during the experiment. Photographic images of the uterus and fetuses of sow W19-117 (**C**) and sow W19-118 (**D**).

**Figure 4 vaccines-11-00054-f004:**
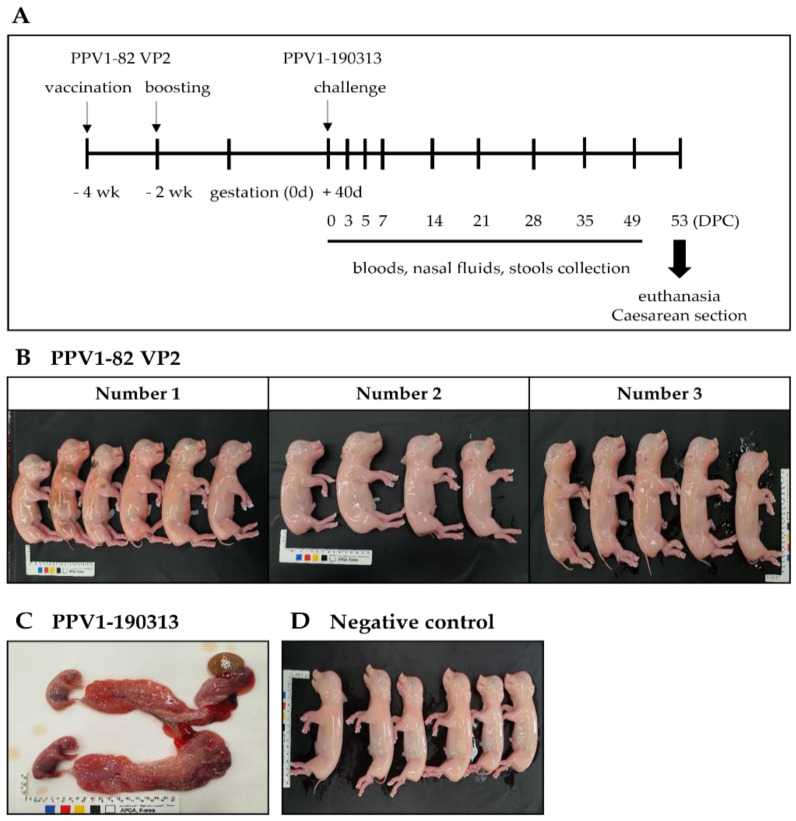
The vaccination and challenge experiment. Sows were intramuscularly vaccinated with the PPV1-82 VP2 antigen (HA value = 2^13^) combined with the MONTANIDE IMS1313 adjuvant at 50% (*v*/*v*) at 4 and 2 weeks before conception. At 40 days after conception, vaccinated sows were challenged with 4 mL of the PPV1-190313 strain (at 4.0 × 10^5^ TCID_50_/mL); 2 mL was administered by intramuscular injection into the neck, and the other 2 mL was delivered intranasally via a spray. (**A**) Schematic representation of the experimental design. (**B**) Fetuses from vaccinated sows. (**C**) The uterus and fetuses from the unvaccinated PPV1-190313-challenged control sow, which miscarried at 8 DPC. (**D**) Fetuses from the unvaccinated unchallenged negative control sow.

**Table 1 vaccines-11-00054-t001:** *PPV1* detection in the blood and nasal fluid samples from sows infected with the PPV1-190313 strain. Blood and nasal fluid samples were collected at the specified number of days postchallenge (DPC) and analyzed for *PPV1* presence by conventional PCRs; +, *PPV1*-positive; −, *PPV1*-negative.

Sample	Sow	DPC									
		0	3	5	7	14	21	28	35	42	49
Blood	W19-090	−	+	+	+	+	+	+	+	+	+
	W19-117	−	+	+	+	+	+	+	+	+	+
	W19-118	−	+	+	+	+	+	+	+	+	−
Nasal fluid	W19-090	−	+	+	+	+	+	+	+	+	+
	W19-117	−	+	+	+	+	+	+	+	+	+
	W19-118	−	+	+	+	+	+	+	+	+	+

**Table 2 vaccines-11-00054-t002:** *PPV1* detection in fetal tissues from sows infected with the PPV1-190313 strain on gestation day 40. Sow W19-090 miscarried at 9 days postchallenge (DPC). Sows were euthanized at 53 DPC, and the fetuses from sows W19-117 and W19-118 were removed by cesarean section. The fetal tissues were liquidized and subjected to conventional PCR; +, *PPV1*-positive; −, *PPV1*-negative. Gray shading indicates that these tissues were not tested because the fetuses were too underdeveloped.

		Tissue						
Sows	Fetuses	Brain	Lung	Heart	Liver	Kidney	Spleen	Umb *
W19-090 (miscarried at 9 DPC)	1	+	+	+				
	2	+	+	+	+			
	3	+	−					
	4	−	+					
W19-117 (euthanized at 53 DPC)	1	−	+	+	+	+	+	−
	2	+	−	+	−	+	−	−
	3	−	+	+	+	+	−	−
	4 (mummy)							
	5 (mummy)							
	6 (mummy)							
W19-118 (euthanized at 53 DPC)	1 (mummy)							
	2 (mummy)							
	3 (mummy)							
	4 (mummy)							
	5 (mummy)							
	6 (mummy)							
	7 (mummy)							
	8 (mummy)							

Umb *, umbilical cord blood.

**Table 3 vaccines-11-00054-t003:** Detection of *PPV1* antigen in sow blood, nasal fluid, and stool samples. The PPV1-82 VP2 vaccinated group (sows 1, 2 and 3) and the PPV1-190313 control sow were challenged at gestation day 40 with the PPV1-190313 strain. Blood, nasal fluid, and stool samples were collected from sows at the indicated days postchallenge (DPC), and conventional PCR was performed to detect *PPV1*; +, *PPV1*-positive; −, *PPV1*-negative.

Sample	Sows		DPC								
	Group	Number	0	3	5	7	14	21	28	35	49
Blood	PPV1-82 VP2	1	−	−	−	−	−	−	−	−	−
		2	−	−	−	−	−	−	−	−	−
		3	−	−	−	−	−	−	−	−	−
	PPV1-190313	1	−	+	+	+	−	−	−	−	−
	Negative control	1	−	−	−	−	−	−	−	−	−
Nasal fluid	PPV1-82 VP2	1	−	−	−	−	−	−	−	−	−
		2	−	−	−	−	−	−	−	−	−
		3	−	−	−	−	−	−	−	−	−
	PPV1-190313	1	−	+	+	−	+	−	−	−	−
	Negative control	1	−	−	−	−	−	−	−	−	−
Stool	PPV1-82 VP2	1	−	−	−	−	−	−	−	−	−
		2	−	+	−	−	−	−	−	−	−
		3	−	−	−	−	−	−	−	−	−
	PPV1-190313	1	−	+	+	+	−	−	−	−	−
	Negative control	2	−	−	−	−	−	−	−	−	−

**Table 4 vaccines-11-00054-t004:** Detection of *PPV1* in tissues from pregnant sows. Pregnant sows were euthanized at 53 days postchallenge (DPC), and their tissues were collected for conventional PCR analysis; +, *PPV1*-positive; −, *PPV1*-negative.

Sows		Tissue						
Group	Number	SMLN *	Heart	Lung	Liver	Kidney	Spleen	Brain
PPV1-82 VP2	1	−	−	−	−	−	−	−
	2	−	−	−	−	−	−	−
	3	−	−	−	−	−	+	−
PPV1-190313	1	+	+	+	+	+	+	−
Negative control	1	−	−	−	−	−	−	−

SMLN *, submandibular lymph node.

## Data Availability

Data Availability Statements in section “MDPI Research Data Policies” at https://www.mdpi.com/ethics.

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
