# Peer review of "A Plant-Produced Porcine Parvovirus 1-82 VP2 Subunit Vaccine Protects Pregnant Sows against Challenge with a Genetically Heterologous PPV1 Strain"

_vaccines, 2022, doi:10.3390/vaccines11010054_

Round 1
Reviewer 1 Report
Kyou-Nam Cho et al prepared a plant-produced porcine parvovirus VP2 subunit vaccine and induced high levels of HI antibody and neutralizing antibody in the pregnant sow model. The vaccine showed good immune protection effect. This is a very realistic study. However, there are some problems in the manuscript that need to be improved:
1. Line19-20, the NADL-2 strain does not protect animals against genetically heterologous PPV strains. Which PPV1 vaccine is currently used in South Korea? How high is the rate of such immune failure?
2. Line 158-159 PPV1-82 VP2 antigen measured HA value, and should also provide the purity and quality of antigen protein?3. Line 229-239, the current prevalent PPV1 in South Korea is Group E, and the subsequent challenge PPV1 190313 strain also belongs to Group E. Why was VP2 of Group D strain PPV1-82 selected for the preparation of subunit vaccine?
4. Line 263 three piglets, Line 285 pathogenicity of the PPV1-190313 strain three miniature pregnant sows, LINE 263 Pathogenicity of the PPV1-190313 strain And the protective effect of the Line 327-337 PPV1-82 VP2 vaccine for sows. The background information of the pig's origin, breed, age, gestational number of sows and whether it carries other pig viruses is unclear? Please add.
5. In Fig. 4.,Table 3 and 4, there was only one sow in PPV1-190313 control group and one sow in Negative control group. No commercial vaccine group was set up for comparison. It is difficult to obtain pregnant sows with negative PPV1.
Author Response
Response to Reviewer 1 Comments
1. Line19-20, the NADL-2 strain does not protect animals against genetically heterologous PPV strains. Which PPV1 vaccine is currently used in South Korea? How high is the rate of such immune failure?
Response 1. At Line 61-64, I added sentences and two references.
2. Line 158-159 PPV1-82 VP2 antigen measured HA value, and should also provide the purity and quality of antigen protein?3. Line 229-239, the current prevalent PPV1 in South Korea is Group E, and the subsequent challenge PPV1 190313 strain also belongs to Group E. Why was VP2 of Group D strain PPV1-82 selected for the preparation of subunit vaccine?
Response 2. At Line 280-284. we added the sentences.
3. Line 229-239, the current prevalent PPV1 in South Korea is Group E, and the subsequent challenge PPV1 190313 strain also belongs to Group E. Why was VP2 of Group D strain PPV1-82 selected for the preparation of subunit vaccine?
Response 3. At Line 427-430, we added the sentences.
4. Line 263 three piglets, Line 285 pathogenicity of the PPV1-190313 strain three miniature pregnant sows, LINE 263 Pathogenicity of the PPV1-190313 strain And the protective effect of the Line 327-337 PPV1-82 VP2 vaccine for sows. The background information of the pig's origin, breed, age, gestational number of sows and whether it carries other pig viruses is unclear? Please add.
Response 4. At Line 290-292 and 312-315, we added the sentences.
5. In Fig. 4.,Table 3 and 4, there was only one sow in PPV1-190313 control group and one sow in Negative control group. No commercial vaccine group was set up for comparison. It is difficult to obtain pregnant sows with negative PPV1.
Response 5.
As you commented, it is difficult to obtain pregnant Crossbred Pigs with negative PPV1. We used SPF Yukatan miniature sows for the pathogenicity and vaccination experiments. However, it was still difficult to use enough sows with similar gestational days. There was a limit on the number on sows.
Because we used PPV1-190313 strains’ pathogenicity using three sows (Fig 3), we used only one sow in PPV1-190313 control group and one sow in Negative control group in the vaccination experiments. First, when we designed the experiment, we did not consider any commercial vaccine group.
Reviewer 2 Report
In this study, authors evaluated the diversity of the VP2 gene in ten PPV1 strains and developed VP2 protein based PPV1 vaccine using plant expression system.
Major concerns
1. In Figure 1A, it appears that the purified protein is not pure. It shows 3 bands. Authors should have used immunoblotting to confirm which of these bands is VP2 protein. The protein is tagged with 6-His. Why was Ni-affinity not used for purification?
2. In Figure 2D, why is there a significant difference in HI titer between groups at 0 d? Both unvaccinated and VP2 vaccinated groups should have similar HI titer at 0 d.
3. It is not clear how serum neutralization antibody values were determined
Minor comment
1. It is good to address rationale for using plant expression system either in introduction or in discussion.
Author Response
Response to Reviewer 2 Comments
1. In Figure 1A, it appears that the purified protein is not pure. It shows 3 bands. Authors should have used immunoblotting to confirm which of these bands is VP2 protein. The protein is tagged with 6-His. Why was Ni-affinity not used for purification?
Response 1. At Line 280-284, we added the sentences.
2. In Figure 2D, why is there a significant difference in HI titer between groups at 0 d? Both unvaccinated and VP2 vaccinated groups should have similar HI titer at 0 d.
Response 2. At Line 290-292, we added the sentences.
3. It is not clear how serum neutralization antibody values were determined.
Response 3. At Line 235-243, we added the Serum Neutralization paragraph.
Minor comment
1. It is good to address rationale for using plant expression system either in introduction or in discussion.
Response to Minor 1 comment
At Line 79-85, we added the paragraph.
Reviewer 3 Report
Porcine parvovirus (PPV) is a common and important cause of reproductive failure in naïve dams. In the last two decades many novel porcine parvoviruses were described, thus new effective and safe vaccines are needed. In this manuscript, Cho and co-authors develop an effective and safe PPV1 vaccine using a plant expression system. They isolated and sequenced PPV1 VP2 genes from 926 pigs and identified several PPV1 strains for further study. Subsequently, they selected PPV1-82 strain as a vaccine candidate and produced the PPV1-82 VP2 protein in Nicotiana benthamiana. They found that the PPV1-82 VP2 protein could form the virus-like particles and exhibited high agglutination value. Importantly, pregnant sows vaccinated with PPV1-82 VP2 had high neutralizing antibody titers and produced normal fetuses after the pathogenic strain PPV1-190313 challenge, which suggests PPV1-82 VP2 subunit vaccine could be a promising PPV vaccine candidate.
Overall, the manuscript is well written, the methods are described in good detail, and the figures with corresponding legends provide the data in a clear form. The conclusions of the study are supported by the data presented, and are clearly stated. However, minor points need to be improved before the manuscript can be recommended for publication.
Minor comments:
1) Line 34, 5-7 keywords are appropriate
2) Figure legends of fig.3 and fig.4 were poorly written, please make this section informative.
3) Neighbour-Joining (NJ) and maximum-likelihood (ML) methods can apply a model of sequence evolution and are ideal for building a phylogeny using sequence data. These methods are most often used in publications. In this study, why the authors used Maximum Likelihood method for generating the phylogenetic relationship among the selected PPV1 VP2 gene sequences? It is also important to mention the nucleotide substitution model and bootstrap tests employed while generating the tree.
Author Response
Response to Reviewer 3 Comments
1) Line 34, 5-7 keywords are appropriate
Response 1. At Line 34, we added two more key-words.
2) Figure legends of fig.3 and fig.4 were poorly written, please make this section informative.
Response 2. At Line 339-342 and 387-392, we added detailed explains.
3) Neighbour-Joining (NJ) and maximum-likelihood (ML) methods can apply a model of sequence evolution and are ideal for building a phylogeny using sequence data. These methods are most often used in publications. In this study, why the authors used Maximum Likelihood method for generating the phylogenetic relationship among the selected PPV1 VP2 gene sequences? It is also important to mention the nucleotide substitution model and bootstrap tests employed while generating the tree.
Response 3. At Line 261-263, we added some explains.